# Degraded Historical Document Binarization: A Review on Issues, Challenges, Techniques, and Future Directions

**DOI:** 10.3390/jimaging5040048

**Published:** 2019-04-12

**Authors:** Alaa Sulaiman, Khairuddin Omar, Mohammad F. Nasrudin

**Affiliations:** Pattern Recognition Research Group, Centre for Artificial Intelligence Technology, Faculty of Information Science and Technology, Universiti Kebangsaan Malaysia (UKM), Bangi 43600, Selangor, Malaysia

**Keywords:** binarization, document degradation, image enhancement, image quality, accuracy, OCR, image artefacts, computing, image manipulation

## Abstract

In this era of digitization, most hardcopy documents are being transformed into digital formats. In the process of transformation, large quantities of documents are stored and preserved through electronic scanning. These documents are available from various sources such as ancient documentation, old legal records, medical reports, music scores, palm leaf, and reports on security-related issues. In particular, ancient and historical documents are hard to read due to their degradation in terms of low contrast and existence of corrupted artefacts. In recent times, degraded document binarization has been studied widely and several approaches were developed to deal with issues and challenges in document binarization. In this paper, a comprehensive review is conducted on the issues and challenges faced during the image binarization process, followed by insights on various methods used for image binarization. This paper also discusses the advanced methods used for the enhancement of degraded documents that improves the quality of documents during the binarization process. Further discussions are made on the effectiveness and robustness of existing methods, and there is still a scope to develop a hybrid approach that can deal with degraded document binarization more effectively.

## 1. Introduction

There exist varieties of ancient and old historic document collections in a variety of holdings all around the world, each of which are scientifically and culturally significant [1,2]. These documents are of importance, as the same information is not available in recent trends and most times these documents are hand written. Due to their long-term preservations in storage or places where they are kept, documents start to degrade in quality, which makes them difficult to read; manual efforts have to be employed to re-write such documents. In order to maintain the quality of the original content and possess greater accessibility, it becomes necessary to transform such documents into digital form [3]. This is because these documents may often be subject to degradation issues [4].

There are several methods to transform the handwritten documents to digitally compatible text. For example, optical character recognition (OCR) can be used to read text from these documents and store the information in digital text [3]. However, most of the OCR tools are limited by the quality of the document available. Handwritten prehistoric documents, in general, suffer from various degradations. Some of the major factors destroying the legibility of such documents include seepage of ink, uneven illumination, image contrast variation, background “noise”, and other factors. In addition, handwritten text shows different changes in stroke width, stoke connection, and pressure on the surface. Historical documents are also affected by bleed-through, an artefact like water blobs that occurs due to one side of ink on the paper passing through to other side. Moreover, some old content in documents may degrade due to fungus, as these documents have been kept in the same environment for long periods of time. The above issues or artefacts existing in the original documents make it difficult for an OCR system to recognize the text accurately and store them in digital form. In order to deal with these challenges, researchers have widely utilized image processing-based methods to overcome different degradations in documents. The method is referred to as image binarization. Under these conditions, necessary filtering techniques must be introduced to get rid of noise from historical documents entirely and enhance their quality before libraries unveil them for the public to see. The role of binarization is to make a separation between the text that appears on the foreground of the page and the degraded background in document images. In the past, several efforts have been made by scientists to improve binarization schemes and provide a better method for the enhancement of quality of degraded documents in order for an OCR to read and store the correct information in digitized forms. Although several attempts have been made in the past in this domain, degraded document binarization is still an unsolved problem for better accuracy, efficiency, and effectiveness. With the open area of research in this domain, this paper aims to review different issues and challenges in terms of degradation of documents associated with historical documents and the methods that have been implemented to deal with such issues, followed by the future directions towards implementing improved techniques for degraded document binarization. 

## 2. Challenges in Historical Documents

Historical documents are the source of obtaining accurate and meaningful cultural and scientific knowledge that can be used for the information retrieval process. These historical documents are usually available in old libraries and reference centers of several government departments. Often, the importance of these documents has been evidenced with respect to their scientific, legal, and cultural values. Recently, there has been growing interest in transforming these documents into digital forms, which converts them to digitally readable text that requires scanning, extracting text, and storing them in a database. This process involves automatically converting handwritten documents from scanned images to undergo OCR, which takes advantage of recognition methods to extract textual information. However, handwritten prehistoric document in general suffer from various issues referred to as document degradations. These degradations can be corrected with the help of document binarization methods and issues can be resolved in order to extract accurate information using OCR tools. To achieve this, there have been several approaches developed for various document binarization tasks including palm leaf [5], music score [6], parsing floor plan [7], and historical documents. Figure 1 shows the sample source for some of these challenges. 

Although these tasks are inherently the same, these are the same challenges in historical document binarization which makes it a difficult task for accurate binarization. Thus, the main goal of this paper is to review the existing algorithms developed for degraded historical document binarization.

Prior to understanding and review of the degraded binarization techniques, it is extremely important to understand the nature of defects and degradations in historical document. Figure 2 depicts the most frequently seen degraded artefacts in documents, and the following section discusses the nature of defects in details. 

### 2.1. Uneven Illumination

In optical imaging, as incident light gets diminished exponentially along the path of light due to spreading of particles in media, light microscopy images degrade and fall victim to uneven illumination [8,9]. As a combined outcome of background objects, the overlays in fluorescence absorption, and emission spectra, light gets scattered, leading to uneven illumination of microscopy. This leads to difficulties in document image analysis [10]. In the case of the document, when the light condition results in an uneven illumination, recognizing characters/texts from the document image using an OCR results in a severely unacceptable effect; this may lead to various difficulties in efficient document recognition. Figure 3 shows examples of uneven illumination in historical document. Generally, for OCR to provide better accuracy, the process consists of transforming an image from grayscale to binary images and extracting the text. However, due to uneven illumination, binary images have artefacts which result in the wrong extraction of text from artefact regions. 

### 2.2. Contrast Variation

Contrast can be defined as a variation in brightness. Most times, contrast refers to the differences between an object with high intensity and low intensity pixels in an image. The contrast can also be measured in terms of differences between high/low pixels of an object in an image and the background pixel values [11]. Factors such as noisy environment, sunlight, illumination, and occlusion often cause contrast variation, which is extremely non-linear and expressive [12]. Variation in contrast existing within historical and handwritten documents results in the document image-analysis algorithms facing difficulties in terms of applying traditional threshold-based methods to separate the foreground text from the background of document images. One can deal with such issues by applying image enhancement methods prior to performing image binarization. Figure 4 shows an example of contrast variations in historic and hand-written documents. 

### 2.3. Bleed-Through Degradation

Bleed-through or ink bleeding happens when the document is written on two sides of the paper and the ink from one side starts to appear on the other side of the document. When it comes to document binarization, bleed-through degradation poses a major threat. It takes place when ink oozes through one side of the page and spreads across over to the other side, ruining the text there. Many ideas were suggested to counter the issue of bleed-through, and researchers working on this issue faced two major challenges. The first was the difficulty in getting access to degraded documents of high resolution unless in connection with a certain digitization project or library. The second challenge is commonly seen in all restoration techniques, where issues arise at the time of analyzing outcomes quantitatively due to unavailability of actual ground truth [13]. This issue can be solved by either preparing an image of certain degradation quality based on the corresponding ground truth [14] or by forming an image as ground truth by knowing the original degraded images [15]. One can always analyze performance despite not having ground truth by quantifying the way restoration impacts a secondary step, just like the enactment of an OCR system on a document image [16]. Figure 5 shows an example of ink-bleed degradation. 

### 2.4. Faded Ink or Faint Characters

There exists enough historical, public, and political interest when it comes to analyzing a huge range of organizational official papers and transforming them into digital libraries and archives. Most of these official papers were typewritten, which brings forward various issues regarding their recognition [17]. To begin with, every individual glyph (character) inside the document could appear either faint or much stronger than the glyphs around it in opposition to other printed documents. This is directly related to both the form of the original striking head of the exact key as well as the extent of force employed while tying. Secondly, a wide range of typewritten documents continue to last only as carbon copies, etched on an extremely squeaky paper (Japanese paper) possessing a prominent texture. As a result of the machine-driven nature held by the typing process (where there is need to press the key harder in order for the printing to take place on the original and carbon copy), most carbon copies appear blurry [18]. Issues like repeated use and ageing, tears, stains, rust from paperclips, punch holes, disintegration of parts of documents, and discoloration negatively impact historical typewritten documents. Figure 6 shows instances of scanned carbon-copy historical typewritten material with faded ink degradation. 

### 2.5. Smear or Show Through

After the documents have been digitized, more challenges come up in the form of noises and low-resolution components appearing over the documents. Due to this, the visual appearance of the document is negatively affected [19]. There could be various forms of degradations a historical document may be suffering from, all of which were introduced with time and may have differing natures [20]. However, one of the largest issues in documents appears as show-through. Many documents in the past were written on both sides of the paper [21]. The show-through problem appears when ink impressions from one side start to appear on the other side, making the document difficult to read. These documents have to be restored in order to be easily readable. By getting rid of the show-through issue, the image compression time reduces significantly, allowing people to download them faster over the internet. Figure 7 displays an instance of an old degraded historical document with the issue of show-through or smear. A clear background can be obtained if the show-through on this image were to be removed [19]. An example of such degradation is shown in Figure 7. 

### 2.6. Blur

When it comes to document degradation, there are two different types of blurring that appears in documents: Motion blur and out-of-focus blur. In general, motion blur artefacts are caused by the relative speed between the camera and the object, or a sudden rapid movement of camera, whereas out-of-focus blur takes place when light fails to converge in the image. In order to fix the blur issue, the research topics as of late have turned towards the tools for assessing blur in document images to figure out the accuracy of the OCR, hence providing the required response to the user to help them obtain new images in the hopes of getting better OCR outcomes. Some instances of blurring issues in degraded documents is displayed in Figure 8.

### 2.7. Thin or Weak Text

Many times, documents written in the past consist of very thin or weak text. In general, these documents were written with the help of ink or sometimes drawn with paint. Due to the quality of ink or paint used in historic documents, they start to degrade as their ink start to shrink. In other cases, the use of low-quality ink and the nature of paper used also resulted in thin or weak text that makes it difficult to apply binarization methods and extract the text accurately. Researchers, as of late, have been showing more interest in prehistoric document image analysis, which has brought forward various new challenges. Degradation in historical documents such as weak or thin texts have encouraged researchers to come up with enhancement and binarization algorithms good enough to fix these issues [22]. Successive phases in terms of algorithms such as skew detection, recognition, and page or line segmentation were then created for binarized data. An example of thin or weak text is shown in Figure 9. 

### 2.8. Deteriorated Documents

Original paper-based documents could be comprised of various forms of media (such as ink, graphite, water color) and formats (such as rolled maps, spreadsheets, and record books). These documents may well be of great importance as they contain informational, evidential, associational, and intrinsic values [11]. A document consisting of historical, legal, or scientific data is said to have great evidential value as long as the original condition of the media, substrate, format, and images are not radically altered due to modification or deterioration [13]. However, adverse use is not the only medium through which documents face deterioration, loss, and damage [11]. Other aspects such as poor storage, handling and environmental conditions, and inherent instability also contribute. Severe damage and deterioration can also be caused through environmental factors, mainly when it comes to inherently unstable documents. An example of deteriorated document is shown in Figure 10. 

## 3. Handling Historical Document Degradation Issues

Ancient and historical documents, either handwritten or printed, generally suffer from various degradations. Some of the major factors destroying the legibility of such documents include seepage of ink, uneven illumination, image contrast variation, background noise, stroke width, stroke connection, bleed-through and water blobs. There have been several efforts to handle such degradations in historic documents. Contrasts can be defined as a variation in brightness. Most of the time, it acts as the difference between certain objects or features within the image, or among objects and the background [11]. Factors such as noisy environment, sunlight, illumination, and occlusion often cause contrast variation, which is extremely non-linear and expressive [9], in particular, to historical and ancient document contrast and may vary within a scanned image due to uneven illumination. A physics-based method was described in Reference [23] and References [24,25] to recover degraded outdoor scene contrast. Schechner and Karpel [26] introduced a physical model that can handle image degradation. The concept of this approach includes finding scene depth with the help of a polarizer where a number of images are taken with varying orientations. One can deal with image contrast issues by applying contrast enhancement techniques. Lately, a number of researchers came up with new image enhancement models of their own to fix the contrast variation and luminosity issue. An extremely common image enhancement technique is histogram equalization (HE) [27]. This method is usually considered for image enhancement due to the better performance and simplicity it comes with, regardless of image type. Many researchers have also stressed the need for improving histogram equalization-based contrast enhancement such as adaptive histogram equalization that allows one to locally enhance contrast [28]. The fact that this technique treats the image globally makes it quite special [27]. The method also helps in the case of images with foregrounds and backgrounds which may be either dark or bright in degraded documents. 

When it comes to document binarization, bleed-through degradation poses a major threat. It takes place when ink oozes through one side of the page and spreads across to the other side, ruining the text there. Many ideas were suggested to counter the issue of bleed-through, and researchers working on this issue face two major challenges. The first would be the difficulty in accessing high-resolution degraded images unless in connection with a certain digitization project or library. The second challenge is commonly seen in all restoration techniques, where issues arise at the time of analyzing outcomes quantitatively due to unavailability of actual ground truth [16]. This issue can be solved by either preparing an image of certain degradation quality based or the corresponding ground truth [14] or by forming an image as ground truth by knowing the original degraded images [15]. Further, one can always analyze performance despite not having ground truth by quantifying the way restoration impacts a secondary step, just like the enactment of an OCR system on document image [16]. 

A majority of older works [15,29,30] are inclined towards classifying an image based on the pixels available in the background of the image, pixels available in foreground of the image, and pixels spread in the regions of ink bleed. The common aspect across all of these approaches is that the pixels marked as ink-bleed have the same color of pixels as in the document background, which is approximated by calculating the averages of the chosen background pixels. This type of binary classification leads to a wrong interpretation by the algorithm by classifying all the pixels as background. Rather than going for this discrete classification, a better idea is to figure out the amount of ink, background, and intensity of ink-bleed at each pixel [14]. Promising progress has been made in automatic restoration of degraded documents during the last several years. A majority of work focuses on extracting the foreground strokes in terms of actual text and removing the ink-bleed from the document [31]. These research works can be categorized into two categories, where one can deal with the issues with a single side while the other can deal with these issues in terms of utilizing both sides of an image to recover as much information as possible. The older works carried out in this field [24,25,32] have demonstrated various issues in terms of segmentation while dealing with single-sided images. Several methods based on the local and global thresholding have been applied and resulted in efficient performances in the case of cleaner images with less ink-bleed artefacts. Xiaojun et al. [33] came up with an approach towards complex ink-bleed via adaptive thresholding, wherein they first lowered the dimensionality of a colored image (RGB) with the help of principal component analysis (PCA), after which classification can be carried out through an iterative clustering of PCA data in two different groups. In the case of non-blind separation, wherein images from both sides of the document are utilized, an initial alignment of these images is needed. 

Wavelet-based approaches were used [34,35] to enhance the thresholding of ink-bleed pixels. Global alignment was utilized in both of these approaches to determine the corresponding pixels of the two images, and hence led to an imperfect overlay of both images. Ink-bleed was also successfully reduced by Gatos et al. [36] and Moghaddam et al. [37,38] with the help of three diffusion models, one for verso-foreground, one for recto-foreground, and the last one for background. Regarding background information, real-paper scan was used with no text, while the model depended on five user-tuned parameters. In other work, research done by Leedham et al. [15] and Baird et al. [4] was further extended by allowing users to provide mark-up whenever required. With the help of user-assistance, they came out with great results, even for the difficult instances of strong ink-bleed. In another study, the front and back side of ink-bleed document are used to extract information by integrating the Chan-Vese active contour model. This is followed by using the functional minimization method in order to remove artefacts. The method is then applied to restore information in terms of broken foreground strokes back into the document. This approach was able to solve issues of removal of ink-bleed from a document but failed to label the verso and recto images concurrently. 

A wide range of typewritten documents continue to last only as carbon copies of originals, etched on an extremely thin paper (Japanese paper) possessing a prominent texture. As a result of the machine-driven nature, most text appears a blur over characters on the carbon copy [18]. The modern commercial OCR systems do not identify most of the characters in such documents. These are some of the well-known challenges faced by an OCR system. Few research works in the literature have reported the binarization of typewritten documents, and those available emphasized dealing with degraded typewritten documents prior to the OCR process in terms of their enhancement. An attempt was made by Cannon et al. [39] to enhance bi-level images and emphasize artefacts possessing a different nature compared to those seen on degraded historical documents. Antonacopoulos and Karatzas [40] came up with a study in the field of challenging historical documents, studying the impacts of various binarization techniques at several segmentation levels. A new text recovery approach was established by Reference [41] mainly for typewritten documents. Both the above approaches were based on commercial systems for character recognition. A new framework was hence developed by Reference [42,43] for non-commercial uses. 

After the documents have been digitized, more challenges come up in the form of noises and low-resolution components appearing over the documents. Due to this, the visual appearance of the document is negatively affected [19]. However, one of the largest issues when it comes to creating digital records of these prehistoric documents is the show-through. The show-through problem appears when low-quality ink impressions available in one side of the document start to pass through and appear on the other side, making the document difficult to read. A clear background can be obtained if the show-through on this image were to be removed [19]. One could apply a mechanism to improve the contrast of an image that can also help in improving the blur edges of the sketch, text, or the image as a whole. There have been various techniques suggested over the years to get rid of noise from historical documents. However, these techniques work only as long as the document is one-sided or if the show-through is really pale [20]. Kitadai et al. [44] came up with an adaptive wavelet thresholding to help with image compression and de-noising to deal with show-through. A newly developed system created [20] for degraded old documents was quite capable of dealing with degradations occurring as a result of noise, low contrast, non-uniform illumination, or shadows. A novel method for document enhancement was to bring together two of the latest efficient noise-reduction methods [20]. Xu et al. [21] invented a technique that performed anisotropic morphological dilation with the help of implicit smoothing in order to restore degraded character shapes of binarized images. Making the most out of the idea of geodesic morphology that states the binary image and its distance-transformed image are always inter-convertible [44], they were able to apply a smoothing technique to the distance-transformed image rather than the binary image, and then re-convert the same with the help of binarization. Quraishi and Mallika De in their work [19] also came up with a novel approach to improve ancient historical documents with a two-way approach. 

Wu et al. [45] saw it best to use machine-learning approaches to estimate blur and saw a fair bit of success in doing so. Eventually, Shoa et al. [46] suggested a revised feature-learning approach that helps with learning features related to document image blur automatically. To get rid of this issue, various image de-blurring methods were proposed. However, restoring the blurred image has continued to be a challenge due to the non-linear blur motion doing its part to make this issue heavily under-constrained. There are various de-blurring techniques mentioned in the literature [47,48,49], all of which have proven to be effective on natural images, but completely the opposite on degraded document images that contain text, because of the ringing artefacts that emerge due to the de-blurring process. Wu et al. [45] came up with a method that can deal with the de-blurring existing in degraded documents containing text information and this was further embedded into OCR to improve the accuracy; however, the results that surfaced later were not good enough due to the fact that the ringing effect led to image degradation. That said, restoration of blurred document images continues to stand as a difficult challenge. This has been implemented in various studies [47,48,49] that are using Sauvola binarization algorithms on documents containing images with blurred text lines. 

Rooms [48] serves as an instance of a segment-based OCR system that divides the lines into character hypotheses, after which it applies a shape-based geometric classifier across the outline of every character hypothesis to identity the same. However, a very accurate segmentation of characters is needed for such segmentation-based approaches, as inaccuracy could lead to an unreliable OCR output. Besides, the quality of binarization steps also matters in the segmentation process. That said, segmentation issues resulting from failures in binarization turn into low-resolution characters. In such cases, to train on out-of-the-way images having very low resolution, a method that utilizes generative learning is suggested. Character segmentation is carried out at the time of recognition based on features taken out of the characters and spaces between each character. To solve the issue of accurate segmentation, various approaches based on the hidden Markov model [50] suggested that one can successfully use segmentation-free methods as in the case of scanning neural networks, and both would improve the accuracy of OCR [50]. During the last few years, recurrent neural networks (RNN) have been widely used for the prediction and classification of time series data. In a recent work [51], recurrent neural networks (RNN)-based methods called long short-term memory (LSTM) neural networks are used to solve the degraded document through classification of background and foreground pixels. These networks have proven to be more decisive. In various cases of old documents such as Latin scripts, Asian scripts, and text-to-speech recognition, LSTMs have been efficiently used and provide better outcomes [51]. This success achieved by LSTM networks convinced us to conduct further research into the application of LSTM networks for recognizing blurred document images. To ensure that it no longer requires binarization or image de-blurring for blurred documents, LSTM networks were employed directly on the grayscale images [51]. 

Researchers as of late have been showing more interest in pre-historic document binarization, which has brought forward various challenges. Issues in historical scripts such as weak or thin texts have encouraged researchers to come up with enhancement and binarization algorithms good enough to fix these issues [22,52]. As an advanced method, various algorithms as subsequent steps are developed. These algorithms are mainly focused on skew detection, page segmentation, line segmentation, and recognition of binarized data. In a recent work, a text-line segmentation method was developed that can segment strong line text from the hand-written document images, old scripts, and documents containing different degraded documents, and this method is applied on grayscale images directly instead of converting them to binary images. These types of documents may consist of curved text lines, making them hard to binarize. A majority of the current text-line segmentation techniques work only with binary images [53] and not much has been done for grayscale image segmentation [54]. So far, the best available technique for text-line segmentation is that which uses local projection profile in an image, based on their orientation. This is one of the fastest algorithms that has the capability to perform line segmentation in documents with lines of varying moderate skew angles. The algorithm is then extended in order to allow for any of the skew angles by adapting to the skew of every line within the document as it goes on. This approach was able to achieve extremely accurate outcomes on a group of documents that contains degradations in various forms with varying skew.

In summary, there have been many attempts to deal with degraded document and research is still being carried out to overcome the degraded issues existing with historical/ancient documents in both handwritten and printed forms. Although, a number of research studies have been conducted to deal with degradation partially in separate studies, there is an emerging scope to solve different types of degradation using a single method through image binarization approaches. The next section of this article is thus focused on the importance of document binarization, followed by the different schemes employed in document binarization. 

## 4. Document Binarization

In the process of binarization, an approach is followed where some object available in an image is separated out from the background of an image, which is referred to as thresholding. In general, the objects in a degraded document may be presented in terms of characters, graphemes, words, graphics, etc. The principal of thresholding works based on the fundamental idea of converting any grayscale image into a binary image prior to processing it for further steps. This helps in reducing the size of data that can be treated lightly during any computation process. When it comes to degraded documents, most image binarization methods face difficulties due to the issues discussed in Section 2 of this article. A typical binarization method involves performing pre-processing on document images followed by applying a thresholding/filtering approach to obtain a clear image that can further be sent to an OCR system for reading in order to extract the information in digital form, as shown in Figure 11. 

Binarization is the step that is actually performed prior to performing OCR. The aim of binarization is to separate foreground text from the background of a document. The text belongs to the foreground and the motive is to recover the text from the degraded document images. Thresholding is an important parameter that refers to the conversion of grayscale image to a binary image. The thresholding of degraded documents is a major challenge. In recent times, researchers have focused on the development of algorithms for document image binarization, and have used various image-processing approaches to deal with issues discussed in Section 2 of this article. In the sub-sequent sections of this article, methods for degraded document binarization are discussed with a concrete focus towards the issues associated with ancient and historical document in terms of their degradation.

### 4.1. Degraded Document Binarization Methods

In the past, many methods have been proposed to address binarization of degraded documents that usually happens due to different artefacts/defects, as discussed in Section 2 of this article. While looking at the trends in image binarization, they can broadly be classified as global, local, and hybrid thresholding approaches. The methods developed using local and global approaches have both been extensively discussed in various articles focused on degraded image binarization. In recent times, not only the global and local, but also the hybrid thresholding, takes advantage of both local and global thresholding as trending topics. The subsequent sections of this article thus discuss the binarization approaches. 

#### 4.1.1. Global Thresholding-Based Binarization Methods

The term global thresholding refers to an approach of image thresholding where a single threshold value is set for the whole image. In thresholding, the pixels are classified into groups and considered for its inclusion and exclusion from an output image based on the set threshold limit. In this case, for an example of a grayscale image, first, all intensity values are calculated, and a global value is marked to be the threshold that is usually calculated based on the histogram of signal intensity in the image; then, a comparison is made where the pixel values above/less than that are either included or excluded from the resultant image. The final output is then seen as the location of pixels as a region of interest in an image with the values based on the calculated threshold. One of the most common and widely used methods for global thresholding is Otsu’s method [55]. Similar to Otsu, Kittler and Illingworth [56] have introduced a method based on thresholding which can efficiently to extract the text from the background. However, these global thresholding methods work efficiently in the case of a rather simple and decent-quality document but fail to deliver the accurate quality when documents exhibit degradation such as uneven illumination or noisy background. Figure 12 shows an example of Otsu global thresholding applied on a degraded document. 

#### 4.1.2. Local/Adaptive Thresholding-Based Binarization Methods 

Unlike global thresholding methods, where a single threshold is calculated for a whole image, local/adaptive thresholding phenomenon works based on the calculation of threshold either from each pixel or a set of pixels, which usually depends on certain pixel limits in a given object of an image. An image with multiple objects of similar pixels can be categorized into different classes and a set of thresholds can be calculated for each group of pixels locally in the image. One of the finest examples of local/adaptive threshold was introduced by Bernsen [57], where a local threshold was calculated in an image based on the histogram. Here, the image was classified into groups of pixels with maximum (max (i, j)) and minimum (min (i, j)) intensities and an estimation of mean value was calculated, followed by the set threshold based on the local neighborhood window centered at pixel (i, j). In this case, contrast is also calculated to estimate the difference between pixels with maximum (max (i, j)) and minimum (min (i, j)) intensities and a local contrast threshold (for an example k = 20) is set for both ranges. If the difference is lower than the threshold (k = 20), the group of pixels with these values can be set to one class and the remaining to another class, either as foreground or background. One can assume that since the contrast difference is calculated with this method, suitability of such methods is appropriate in the case of bigger contrast values. 

For the degraded documents, Niblack introduced an algorithm [58] which estimates a threshold based on each pixel in an image (usually a rectangular sliding window) and local mean and standard deviations are taken into consideration. It has been found in the approach proposed by Niblack [58] that, although the method serves to correctly identify text from a degraded image’s document, it also suffers from background noise. In order to deal with issues faced by Niblack’s method, an improved method was then proposed by Sauvola and Pietikainen [59] that usually performed better than Niblack’s method, but often times introduces thin and broken text due to the extensive sliding window operations. The approach was further investigated by Gatos et al. [60], where a method based on multi-stage processing was taken into consideration towards developing a document binarization approach. In this approach, several stages include noise correction and contrast improvement with the help of a Weiner filter, followed by performing segmentation for text from the background based on the method developed by Sauvola and Pietikainen [59]. In the next steps, an intensity analysis was done to identify the background area in an image, which is further followed by generating the final threshold based on the original and resultant image. This threshold is then utilized for final binarization. Figure 13 shows an example of local thresholding outcomes on a degraded document. Although this method performs better for document image binarization, the main disadvantage of the method was that it dealt with only the area with textual information and on the other areas, the method was not showing efficient results. In order to enhance the capacities of Niblack’s methods, another method was proposed by Khurshid et al. [61] that can deal with the issue associated with Niblacks’s method. The proposed method was named Nick’s method. In a more recent study, a novel algorithm was introduced to solve degraded document image binarization by Khan and Mollah [62]. Here, background noise removal and document quality enhancement were performed first, followed by applying the variant of Sauvola’s Binarization method, and ultimately performing post-processing to find small areas of connected components in an image and removing the unnecessary components from the image. 

#### 4.1.3. Hybrid Thresholding-Based Binarization Methods

In the previous sections of this article, degraded document issues were discussed, followed by the global and adaptive thresholding approaches used during the binarization process. The process of local or global thresholding have their own advantages and disadvantages, and one can use the strength of both local and global thresholding methods in order to provide a better mechanism. This concept has been seen as an efficient solution, and researchers have come up with methods combining both the approaches and named the new approach hybrid binarization methods. There are several benefits of hybrid binarization methods, and one can take into consideration the computation time, flexibility and efficiency of methods, robustness, and accuracies in terms of background and foreground region extractions. This approach has been considered by Su et al. [63], where different thresholding methods were combined to provide significantly better outcomes. In this approach, an improvement was carried out by combining global (Otsu’s) and adaptive (Sauvola’s methods) for better image binarization. In the same study, algorithms developed by Gatos’s and Su’s methods were combined along with combining Lu’s and Su’s methods, which resulted in the best approach. Figure 14 shows an example of combining binary methods as proposed by Su et al. [63]. 

In an attempt made by Sokratis et al. [64], a better result was obtained in the final binarization, where both the local and global approaches were combined. In this method, the algorithm used global and local threshold as first iterative global thresholding, followed by the detection of a noisy area, and again, the iterative global thresholding was applied locally in each area with noise. An example of the hybrid algorithm developed by Sokratis [64] is shown in Figure 15. 

Another study employed the use of a novel hybrid binarization where ensemble-of-expert (EoE) and grid-based Sauvola’s methods were combined as an entry point to provide better outcomes [65,66]. In general, an EoE framework is developed where the outputs of several methods are taken into consideration and a confidence map is generated. This confidence map is further utilized to predict the best possible image as an outcome of the general final document image in binarized form. In the next steps, grid-based Sauvola’s technique was employed by adjusting different parameters through grid-based modeling that helps in reducing computation time and memory utilization. As the final step, a texture analysis was performed as post-processing steps that can extract the best possible text based on the texture mechanism and results in obtaining a better binarization of degraded document images. In another work proposed by Mitianoudis and Papamarkos [67], a new algorithm was developed for the improvement in document image binarization and this approach consists of three stages. In the first stage, the background was removed with the help of an iterative median filter, followed by the second step, which helps in the separation of misclassified pixels. For the separation of these misclassified pixels, local co-occurrence mapping (LCM) and Gaussian mixture clustering are used, and finally, morphological operators are applied for the identification and suppression of misclassified pixels which, in general, results in better classification of text and the background image pixels. 

#### 4.1.4. Machine Learning-Based Binarization Methods

Traditional global and local thresholding methods have difficulties in the binarization of images with noisy and non-uniform backgrounds. Machine learning approaches are an alternative way to overcome these difficulties. Many machine learning algorithms have been widely applied to many document binarization tasks. As an initial work, a neural network-based algorithm was proposed by Yan and Wu [68] for character extraction from document images. In this approach, a multilayer neural network is trained with document samples and their related ground truth for pixel-level document classification. The neural network inherently extracts features from pixels by considering their neighborhood and has the capability to produce adaptive binarization results. However, this process requires more training data and proper training time. In order to reduce the training time, a two-stage binarization approach was proposed by Chi et al. [69]. In this method, at first, a region-based binarization stage is applied to the input image to generate the binary output. To do this, the input image is binarized by using global and local threshold values, which were extracted from the input image by calculating the histogram of the entire image and pixel level intensity. Then at the second stage, a neural network is applied on the extracted binary image to distinguish characters from the background.

The multilayer perceptron is a feed-forward structure with high capability in learning complex patterns for the various challenging tasks such as cleaning hand-written data [70] and document binarization [71]. Kefali et al. [72] have proposed feed-forward neural networks for document binarization in old manuscripts. This method relies on the classification scheme using a multilayer perceptron network. In this approach, the document images with their related ground truth were fed into a multilayer perceptron network to learn the sample patterns. At test time, a learned feed-forward network is applied to the input document image to generate the binarization result. The experimentation results demonstrated that the proposed Multilayer perceptron (MLP) network is able to accurately separate foreground information from the degraded documents [73]. 

Although the artificial neural network has a high capability in learning complex pattern recognition tasks, they still have misclassification results on degraded document binarization. Fortunately, deep neural networks, which are a hierarchical NNs, have demonstrated a vast representational capacity and high-performance rate in various document image binarization tasks [74]. Westphal et al. [74] have proposed a recurrent neural network model for document image binarization. In this method, they utilized the grid LSM cells [75] to handle multidimensional input in order to incorporate contextual information in each step of the binarization process. To do so, they divided the input document image into non-overlapping blocks of 64 × 64 pixels. These blocks are then considered as an input sequence for the RNN model. Each block is then read by four separate grid LSTM layers ($L^{1}$) and the output of these layers are combined and aligned to produce the mid-level feature sequence. Then, these mid-level features are fed to two different grid LSTM layer ($L^{2}$), followed by a bi-directional LSTM to produce the high-level feature map. Then, a full connection layer ($L^{3}$) has been applied on the high-level feature map to produce the binarization result. Experimental results demonstrated a significant increase in binarization quality. The evaluation process revealed a trade-off between binarization quality and execution time with considering different choice for scale factor, footprint size, and the loss function. 

In Reference [76], a convolutional auto-encoder decoder model has been proposed for document image binarization. In this method, a hierarchical layer of convolutions followed by down-sampling layers were applied on the input image to generate the mid-level representation. In fact, the purpose of the auto-encoder section is to generate the mid-level representations in order to describe the relationship between parts of the image that was initially far apart. Then, these mid-level representations were followed by a series of convolutional plus up-sampling layers to generate the binarization mask. Once the model is trained, the network output is then filtered by a global threshold to produce the binarization result. The model was evaluated using different strategies and demonstrated remarkable improvement in document image binarization. It is worthwhile to mention that this approach increased the best average binarization performance obtained by the state-of-the-art methods from 75.48 to 83.41 in terms of F-measure.

Tensmeyer and Martinez [77] have proposed a fully convolutional neural network for binarization of both document image and palm leaf manuscript. The model tries to perform document binarization as a pixel level classification task with minimizing the combined P-FM and FM loss functions. More specifically, the model takes both grayscale and locally computed relative darkness [77] features as an input and applies several encoding layers (convolution layer followed by max-poling) to generate multi-level feature maps. These feature maps, which shows different scale information, are then followed by up-sampled layer to generate the same sized high-level feature maps. These high-level feature maps are then concatenated and followed by convolutional and sigmoid function to generate the binarization mask. Finally, by applying a constant threshold on the network, the binarization image is produced. Experimental results on both document image binarization and palm leaf manuscripts showed promising results. In addition, the paper states that using additional feature maps such as a relative darkness feature can increase the model performance on a binarization task.

Ref. [78] proposed a hierarchical deep supervised network (DSN) for document image binarization. The network architecture follows two main goals. The first aim of this network is to use high-level features to distinguish foreground (text) from the noisy background and the second objective is, while dealing with noisy background and some other challenges, to preserve foreground (text) information with high visual quality. In order to do, the network is designed in a way to use different level features for preserving high details of the foreground. The proposed network consists of three different DSNs, in which the first DSN contains a low number of convolution layers to generate the low-level feature maps. The second DSN consists of a slightly deeper structure for producing mid-level feature maps, and the last DSN is a deep structure for generating high-level feature maps. The input of these DSNs are the same grayscale images with binary masks. These three DSNs were trained independently in order to produce foreground maps at three different feature levels. Then, these feature maps are combined to produce the final binarization map, which is followed by a threshold value to produce the binarization result. The experimental result has been done on three public datasets and demonstrated that the proposed model fully outperforms the state-of-the-art binarization algorithms. In addition, the authors evaluated the results of the proposed method on different types of documents, effects of data augmentation, and noisy inputs on model performance. Furthermore, the hierarchical structure of the model demonstrated a significant increase in preserving text strokes and provides excellent visual quality. 

## 5. Performance Metrics and Advanced Document Binarization

Once a binarization method is developed using either of the methods discussed above, the next step is to measure the efficiency in terms of evaluation and comparison. Document image binarization competition (DIBCO) has provided benchmark evaluation metrics that made use of mathematical approach to measure the accurateness of obtained results against the ground truth data available. The next section discusses various evaluation metrics and their importance with respect to measuring the performance of methods for degraded document binarization. The subsequent section then focuses on the various methods developed to deal with existing degradation types in historical/ancient documents, and their performance.

### 5.1. Performance Metrics to Evaluate Binarization Methods

To measure the performance in terms of the efficiency of a document binarization method, results obtained should be evaluated and compared using various metrics; some of the measures suggested by Gatos et al. [79] and Pratikakis et al. [80] are depicted in Figure 16, and their description is provided in subsequent sections. 

#### 5.1.1. F-Measure

This criterion of performance measure [79,80,81,82,83] is calculated in terms of recall and precision, where a combined assessment is carried out to evaluate the overall performance. In this case, harmonics mean is calculated using both precision and recall function as follows given in Equation (1).
(1)$$F-measure=\frac{2\times recall\times precision}{recall+precision}$$
where, recall and precision are denoted as follows
$$Recall=\frac{TP}{TP+FN}$$
$$precision=\frac{TP}{TP+FP}$$
where, *TP* denotes true-positive rate, *FP* represents false-positive rate, and *FN* represents false-negative values.

#### 5.1.2. Pseudo-F-Measure

Pseudo-F-Measure [84] is another way of measuring performance similarly to F-Measure. However, it uses the pseudo function in terms of recall and precision rather than their direct functions. The advantage of using pseudo-recall and pseudo-precision is that they use weighted distance between output images as boundaries of characters in the extracted document and the boundaries of characters in the ground-truth (GT) image. One other advantage of using pseudo nature of recall is its consideration towards local stroke width in output images, while pseudo-recall takes into consideration the local stroke width, and pseudo nature of precision grows to the stroke width of connected component in ground truth images. 

#### 5.1.3. Peak Signal-to-Noise Ratio (PSNR)

Peak signal-to-noise ratio (PSNR) [85] is a measure of the amount of signal in an image with respect to the amount of noise available. The higher values of PSNR relate to better signal as compared to noise in image. With respect to document binarization, PSNR provides a measure of the quality of binarization against ground truth image and it can be measured as given in Equation (2).
(2)$$PSNR=10\times log_{10}\left(\frac{MAX\frac{2}{I}}{MSE}\right)$$
where, mean square error (MSE) can be represented as below.
$$MSE=\frac{\sum_{i=1}^{N}\sum_{j=1}^{M}{\left(B_{\left(i,j\right)}-GT_{\left(i,j\right)}\right)}^{2}}{N\times M}$$

#### 5.1.4. Negative Rate Metric (NRM)

The negative rate metric (NRM) [86] is another performance metric given in DIBCO and it calculates actual mismatches of pixels between the results output and ground truth images based on pixel by pixel estimations. NRM, in general, is a ratio of total false-negative and total false-positive pixels in combination and it can be expressed as in Equation (3). In this case, the lower the NRM, the better quality binarization method
(3)$$NRM=\frac{NR_{FN}+NR_{FP}}{2}$$
where, *NR_FN_* and *NR_FP_* can be calculated as given below
$$NR_{FN}=\frac{N_{FN}}{N_{FN}+N_{TP}}$$$$NR_{FP}=\frac{N_{FP}}{N_{FP}+N_{TN}}$$
where, *N_TP_* is true positives, *N_FP_* is false positives, *N_TN_* is true negatives, and *N_FN_* is false negatives. 

#### 5.1.5. Multi-Classification Penalty Metric (MPM) 

The misclassification penalty metric (MPM) [87] was also used in DIBCO as a performance metric to evaluate the performance of binarization methods. It can predict object-by-object distances and mismatches between the resultant output objects and ground truth objects, and it can be presented as given in Equation (4). For the measure, a low MPM score represents the better quality of the binarization method.
(4)$$\mathrm{MPM}=\frac{MP_{FN}+MP_{FP}}{2}$$
where, *MP_FN_* and *MP_FP_* can be calculated as given below
$$MP_{FN}=\frac{\sum_{i=1}^{N_{FN}}{d_{FN}^{i}}_{.}}{D}$$
$$MP_{FP}=\frac{\sum_{j=1}^{N_{FP}}{d_{FP}^{j}}_{.}}{D}.$$

#### 5.1.6. Distance Reciprocal Distortion (DRD)

Distance reciprocal distortion [86] is a measure that quantifies the distortions of all pixels that have been modified during the binarization process. DRD can be calculated as expressed in Equation (5):(5)$$DRD=\frac{\sum_{k=1}^{N}DRD_{k}}{NUBN}.$$

*DRD_k_* is the reciprocal distortion distance and it can be calculated as given in Equation (6). NUBN is the number of the non-uniform (not all black or white pixels) of the 8 × 8 size in ground truth image.
(6)$$DRD_{k}=\overset{2}{\underset{i=-2}{\sum }}\overset{2}{\underset{j=-2}{\sum }}\left|B_{k}\left(x,y\right)-GT_{k}\left(i,j\right)\right|\times W_{NM}\left(i,j\right)$$

#### 5.1.7. Average Quality Score

Average quality score [86] is another measure based on the statistics that can provide a global sign of a method’s efficiency in a competitive environment. For a comparison between outcomes of developed methods against ground truth values, the average quality score can be expressed as given in Equation (7) below. This method is used to find the best methods among the few that are in comparison, and the lowest accumulated scores represents the best.
(7)$$S_{i,j}=\overset{N}{\underset{k=1}{\sum }}R_{i}\left(j,k\right)$$
where *N* is the number of evaluation criteria; *R_i_*(*j, k*) denotes the individual ranking value. 

## 6. Advances in Binarization Methods

In the past, numerous studies have been conducted to deal with document binarization of handwritten as well as printed documents with different types of degradations [88,89,90,91,92,93,94,95,96,97,98]. The degradation in handwritten and printed historical documents exists in different forms, and a detailed discussion on these issues has been made in Section 2 of this review. Since there are multiple types of degradations available, our review focuses on the approaches that deal with such degradations and compare the results obtained. Table 1 below includes different studies conducted in the past to deal with various kinds of degradations, and their outcomes and performances are included to compare the availability of the best methods in the literature.

The above studies listed in Table 1 are conducted from time to time and each subsequent method is developed with an intention to improve the results. For the methods developed for degraded document binarization, performance is measured by evaluating various metrics discussed in the previous section of this article. Another important aspect of the developed method was to evaluate their performance on different databases available. Mainly, DIBCO document images are extensively used as a benchmark. Looking at the studies listed in Table 1, several methods have achieved significant performance. However, these methods have been implemented to solve one or a few document degradation issues discussed in the Section 2 of this article. Some methods are better than others for a specific issue, but there is no global method that can deal with all issues related to document degradation. There is a compelling need for the development of globally accepted method that can provide significantly robust results to deal with different kinds of document degradations.

## 7. Conclusions

In this review, different document degradation issues are discussed. These issues are associated with ancient and historical documents, either in handwritten or printed form. The existence of degradation in handwritten and printed documents exhibits several challenges in developing an accurate and robust method for document binarization, which ultimately aims at improving the quality of different OCR systems. In this review, different binarization techniques are discussed that are recently on trend. There are so many binarization methods available that can work well with a particular type of degradation, but a binarization method that can handle any type of degradation is still left for future work. Binarization is an important step towards the development of a system for document image recognition and it has a wider application in the current era in which everything is moving towards digitization. Although several efforts have been made in the past to fulfill the main objective of binarization that involves separation of text from the degraded document background, there is a compelling need to develop a fast and accurate method of binarization followed by the OCR system. The futurist approach should focus on the time and accuracy along with the global method to deal with various kinds of issues associated with prehistoric documents. Moving forward, one can focus on developing a method that can resolve the issues of degraded document followed by combining various machine learning-based methods to automatically identify and select the suitable methods with respect to the degradation available in the document. Towards this, there is also a need for machine learning-based methods to classify the degradation issues automatically from the document. In the future, one can improve the quality of degraded document prior to performing binarization and in doing so, an image enhancement method can certainly help in the future. 

## Figures and Tables

**Figure 1 jimaging-05-00048-f001:**
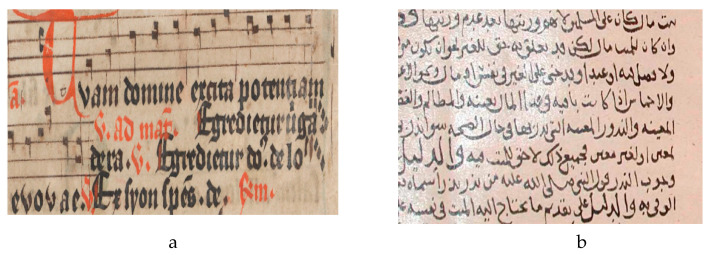
Source sample of different degraded documents; (**a**) music score [6], (**b**) historical document [8].

**Figure 2 jimaging-05-00048-f002:**
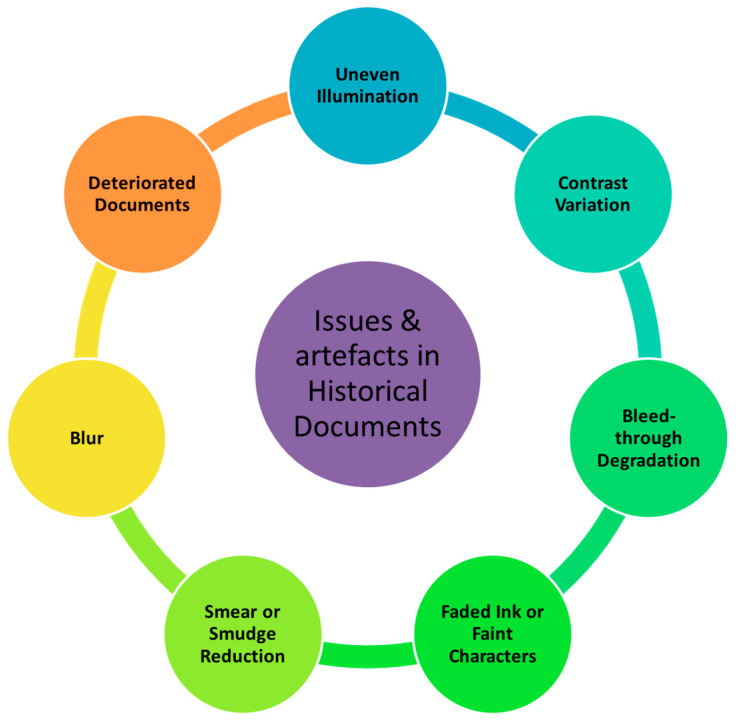
Most frequently seen degraded defects in historical documents.

**Figure 3 jimaging-05-00048-f003:**
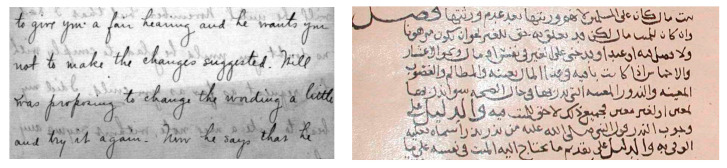
Uneven illumination in historical handwritten document from DIBCO (**left**) and Arabic Databases (**right**).

**Figure 4 jimaging-05-00048-f004:**
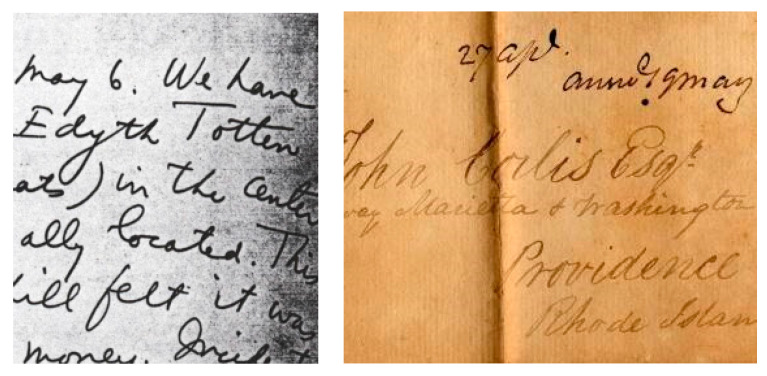
Degraded document images showing variation of contrast: Left image shows higher contrast at the left edge while lower contrast is shown at the right edge of image. The right image depicts low contrast in the middle and high contrast in both left and right edges.

**Figure 5 jimaging-05-00048-f005:**
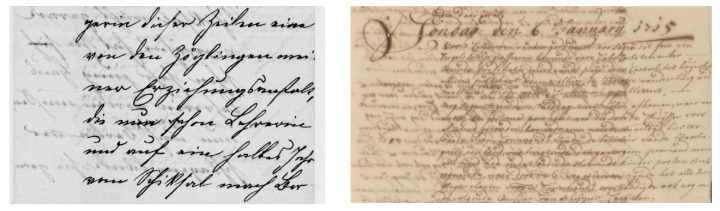
Example of ink-bleed degradation in handwritten documents from DIBCO database (**left**) from 2016 and (**right**) from 2017 databases.

**Figure 6 jimaging-05-00048-f006:**

Example of DIBCO (**left**) and Arabic (**right**) database images showing faded degradation.

**Figure 7 jimaging-05-00048-f007:**
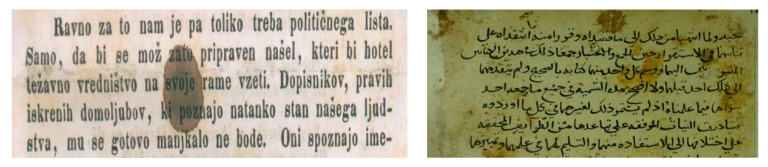
Example showing degraded document with smear/show-through effects.

**Figure 8 jimaging-05-00048-f008:**
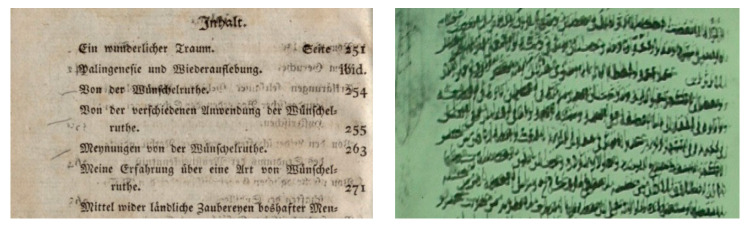
Example showing degraded document with blurring effect.

**Figure 9 jimaging-05-00048-f009:**
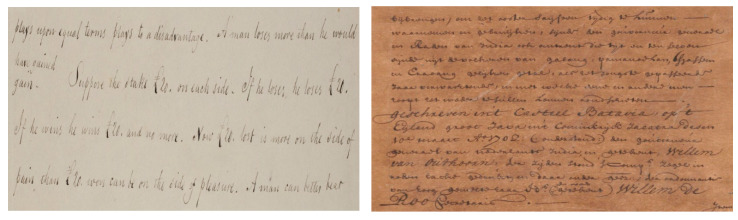
Example showing thin or weak text from old document.

**Figure 10 jimaging-05-00048-f010:**
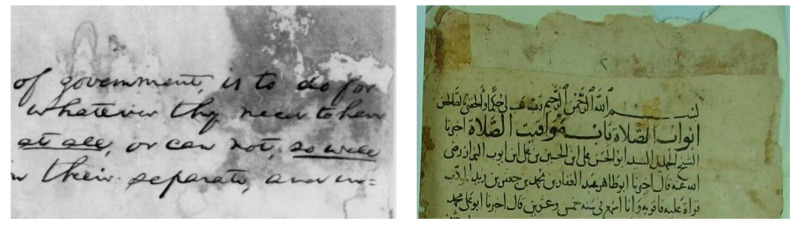
An example of deteriorated documents from DIBCO and Arabic databases.

**Figure 11 jimaging-05-00048-f011:**
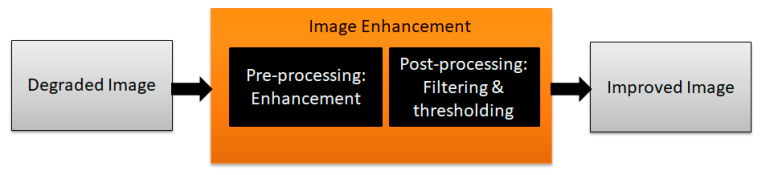
Typical architecture of degraded document binarization.

**Figure 12 jimaging-05-00048-f012:**
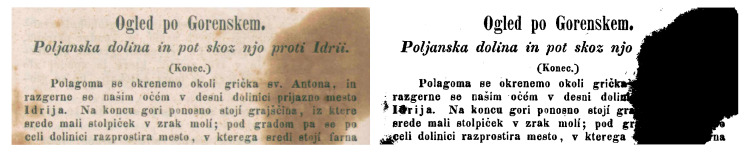
Example of Otsu-based binarization that fails to solve the problem of degradation.

**Figure 13 jimaging-05-00048-f013:**
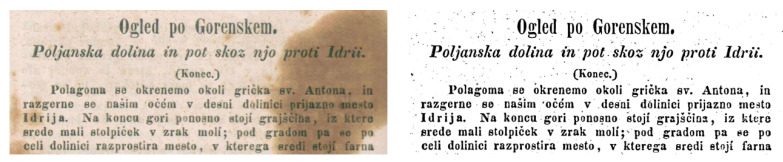
Example of adaptive thresholding applied on a degraded document image.

**Figure 14 jimaging-05-00048-f014:**
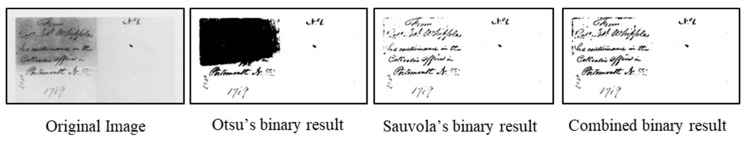
Example showing results obtained from Otsu and combined binary approach.

**Figure 15 jimaging-05-00048-f015:**
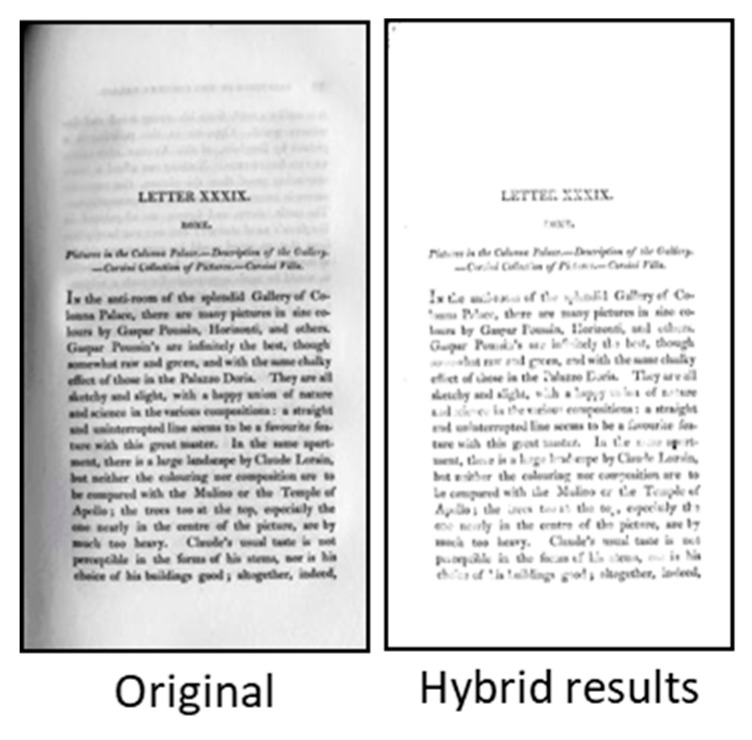
Example showing the results obtained using hybrid binarization method proposed by Sokratis et al. [64].

**Figure 16 jimaging-05-00048-f016:**
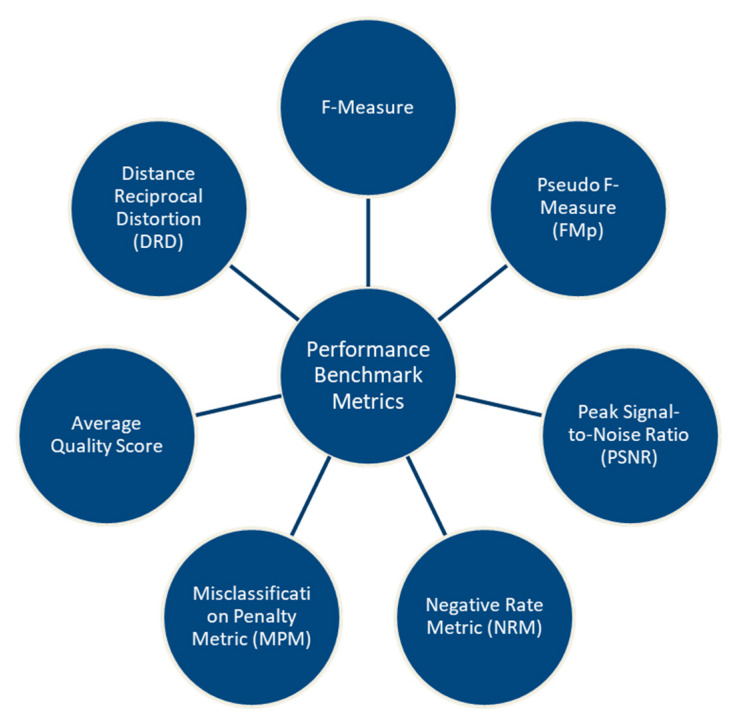
Benchmark performance metrics for document binarization methods.

**Table 1 jimaging-05-00048-t001:** Comparison of popular document binarization methods for degraded documents.

Studies Conducted	Outcomes/Performance
Niblack, 1985 [59]	Performance is measured in terms of distance calculated from results and ground truth. The distance was 146.
Li and Lee, 1993 [88]	Average quality score was measured as 0.114 which shows good performance
Sauvola et al., 1997 [89]	Performance is measured in terms of distance calculated from results and ground truth. The distance was 28.7.
Cheng et al., 1998 [90]	Qualitative measure shows preservation of main features of the component image
Sauvola et al., 2000 [60]	Performance in evaluated based on the ranking/scoring in terms of weighted distance from base pixels. Overall weighted performance was 93.2% as compared to other methods that results in less than 90%.
Wolf et al., 2002 [91]	Performance is measured in terms of distance calculated from results and ground truth. The distance was 53.16.
Kavallieratou and Stathis, 2006 [92]	Performance in measured in terms of precision from 43 good-quality images that shows high values as 97.67%
Gatos et al., 2009 [79]	Performance in measured in terms of calculating F-measure and values obtained are 91.9%
Kuo et al., 2010 [92]	Performance in visually measured and results shows competitive visual result compare to Niblack, Sauvola, Chang, and Otsu methods
Lu et al., 2010 [94]	Performance in measured in terms of calculating F-measure and values obtained are 91.24%
Pai et al., 2010 [95]	Performance is measured in terms of time and recognition rates and the values obtained are:Average processing time = 0.351; Average recognition rate = 98.22%
Bataineh et al., 2011 [96]	F-mean, PSNR, NRM respectively are 10.5, 6.16 and 89.34
Neves and Mello, 2011 [98]	Values obtained for F-measure, PSNR, NRM, MPM, and GA are 88.7052, 18.7090, 0.0576, 0.6823, and 0.9360, respectively
Su et al., 2011 [64]	F-Measure was calculated and combined results of Otsu’s and Sauvola’s are 86.62% while combined results of Lu’s method and Su’s are 93.18%
Singh et al., 2011 [99]	Computational Time is measured as ~0.234 Sec. Achieved lowest computational time compared Niblack, Sauvola, and Bernsen methods
Moghaddam and Cheriet, 2012 [100]	Using the F-measure method automatic mode = 91.57%. Grid-based AdOtsu = 92.01%. Multiscale grid-based AdOtsu = 92.06%
Howe, 2013 [97]	Performance is measured in terms of F-measure (%) PSNR and DRD are 89.47, 21.80, and 3.44, respectively.
Kefali et al.2014 [72]	Performance increase according to PSNR, DRD, etc. Metrics.
Ntirogiannis et al. 2014 [101]	F-measure, PSNR, NRM, MPM, p-F-measure (p-FM) and Distance Reciprocal Distortion (DRD), top performance in most cases.
Hadjadj et al. 2014 [13]	F-measure method = 91.24%
Mitianoudia and Papamarkos, 2015 [67]	Performance in measured by calculating PSNR, MSE, Recall, Precision, F-mean, NRM with the values obtained as 15.29, 0.0295, 0.8331, 0.9466, 0.8862, and 0.0872, respectively
Al-Khatatneh et al. 2015 [102]	Performance of this method is measured for handwritten and printed documents. For handwritten document, F-mean and PSNR are 79.63% and 16.56 respectively while for printed document F-mean and PSNR are 87.6% and 15.94 respectively
Lu et al., 2016 [103]	This method was evaluated in terms of F-mean, PSNR and NRM for both the printed and handwritten document. For handwritten images, F-mean, PSNR and NRM are 82.82%, 11.86, and 8.76, respectively while for printed documents values are 87.12%, 10.44, and 8.92, respectively
Calvo-Zaragoza et al. 2017 [76]	Remarkable accuracy rate achieved by the deep Binarization algorithm. 7.9% increase in the accuracy rate vs. state-of-the-art method
Bataineh et al., 2017 [104]	Measured F-mean, PSNR, NRM and obtained the results as 17.5, 5.14, and 88%, respectively
Tensmeyer et al. 2017 [77]	Performance is measured with various metrics demonstrated an improvement in final performance on two public datasets
Chen et al., 2017 [105]	The performance in measured in terms of PSNR and F-mean that shows the values as 18.2381 and 95.7, respectively.
Hadjadj et al. 2017 [106]	Performance is measured in terms of F-measure (%) PSNR and DRD are 91.67, 19.96 and 2.76 respectively.
Westphal et al.2018 [74]	Performance increase in terms of PSNR and BRD metrics against state-of-the-art models
Quang Nhat Vo et al. 2018 [78]	Significant improvement achieved by evaluating different metrics: F-measure 94.4, PSNR 21.4, BRD 1.8
Lu et al. 2018 [107]	Performance in measured in terms of FP, FN, TP, TN, FP + FN, F-measure (%) and NRM and values obtained are 3180, 8968, 37,013, 747,992, 12,148, 85.90%, and 0.0996 respectively that shows better performance as compared to Otsu, Niblack and Sauvola methods.
Khitas et al. 2018 [108]	Best Results obtained on BICKLEY DIARY Dataset. Values obtained for the F*m* (%) PSNR, NRM and MPM are 79.11, 13.24, 12.85, and 23.99, respectively. This method has achieved comparatively less significant results for DIBCO database
Xiong et al. 2018 [109]	Superior performance in terms of F-measure, PSNR, NRM, DRD, and MPM that is obtained as 89.967, 18.640, 0.054, 3.757, and 1.979 respectively that shows far better performance
Boudraa et al. 2019 [110]	They used terms of FM (%), FM*p* (%), DRD and PSNR are 85.08, 89.81, 5.08, and 17.47 respectively

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
