# Peer review of "Degraded Historical Document Binarization: A Review on Issues, Challenges, Techniques, and Future Directions"

_2313-433X, 2019, doi:10.3390/jimaging5040048_

Reviewer 1 Report

The paper discusses the approaches on binarization of degraded document images. However, it does not contain recent approaches based on deep neural networks, which show significantly improved performance. This is a survery paper without including most recent and most effective methods. The paper should include such approaches and should be submitted again.

Author Response

Dear  reviewer,

 We sincerely would like to thank the respected anonymous reviewers for their constructive and precise comments that led to improve the quality of the manuscript. In the revised manuscript, we have done our best to address all raised comments. Responses to the technical comments are listed below and the corresponding changes have been applied to the revised manuscript.

Reviewer 2 Report

This paper surveys the task of degraded document binarization. The paper introduces the problem, reviews the main challenges, and discusses the existing techniques.

In favor of the paper, I liked the idea of including Section 2, but I think the title could be more information if renamed to something like 'Challenges in Historical Documents). Also, most parts of it are really comprehensive. However, although the topic is interesting for a broad audience of this journal, I feel the paper still lacks content to be published as a proper review.

My main concerns are as follows:

- I think the paper is too focused on binarization for text paper documents. However, there are much more contexts in which binarization is necessary and/or used, such as palm leaf manuscripts [1], music scores [2], or floor plannings [3]. The paper should mention these contexts, and as many as possible, as well.

[1] J.-C. Burie, M. Coustaty, S. Hadi, M. W. A. Kesiman, J.-M. Ogier, E. Paulus, K. Sok, I. M. G. Sunarya, D. Valy, ICFHR2016 competition on the analysis of handwritten text in images of balinese palm leaf manuscripts, in: Frontiers in Handwriting Recognition (ICFHR), 2016 15th International Conference on, IEEE, 2016, pp. 596–601

[2] J. Calvo-Zaragoza, G. Vigliensoni, I. Fujinaga, Pixel-wise binarization of musical documents with convolutional neural networks, in: Fifteenth IAPR International Conference on Machine Vision Applications, 2017, pp. 362– 365.

[3] Dodge, Samuel, Jiu Xu, and Björn Stenger. "Parsing floor plan images."  in: Fifteenth IAPR International Conference on Machine Vision Applications, 2017, pp. 358--361

- The survey completely misses machine learning-based methods for document binarization. These might deserve a (sub)section on their own. There are many like:

-- Z. Chi, K. W. Wong, A two-stage binarization approach for document images, in: Proceedings of the International Symposium on Intelligent Multimedia, Video and Speech Processing, IEEE, 2001, pp. 275–278. 23 

-- J. L. Hidalgo, S. Espana, M. J. Castro, J. A. Perez, Enhancement and cleaning of handwritten data by using neural networks, in: Proceedings of the 2nd Iberian Conference on Pattern Recognition and Image Analysis, Springer Berlin Heidelberg, Berlin, Heidelberg, 2005, pp. 376–383. 

-- A. Kefali, T. Sari, H. Bahi, Foreground-background separation by feed forward neural networks in old manuscripts, Informatica 38 (4).

Anyway, in my opinion, it is really important to consider those that make use of Deep Neural Networks, as they represent the state of the art in many image-processing duties, like:

--  J. Pastor-Pellicer, S. E. Boquera, F. Zamora-Mart´ınez, M. Z. Afzal, M. J. C. Bleda, Insights on the use of convolutional neural networks for document image binarization, in: Advances in Computational Intelligence - 13th International Work-Conference on Artificial Neural Networks, IWANN 2015, Palma de Mallorca, Spain, June 10-12, 2015. Proceedings, Part II, 2015, pp. 115–126.

-- X. Peng, H. Cao, P. Natarajan, Using convolutional encoder-decoder for document image binarization, in: 14th IAPR International Conference on Document Analysis and Recognition, Kyoto, Japan, 2017, pp. 708–713.

-- Jorge Calvo-Zaragoza, Antonio-Javier Gallego: A selectional auto-encoder approach for document image binarization. Pattern Recognition 86: 37-47 (2019)

Taking into account the nature of the paper, I consider these modifications necessary to consider the manuscript as a proper survey on document binarization.

Other minor comments:

- L51 is misleading: image binarization is not the only task to overcome document degradation (bleed-trough removal, slant correction, light equalization, etc).

- Some careful English proofreading is needed.

Author Response

Dear  reviewer,

 We sincerely would like to thank the respected anonymous reviewers for their constructive and precise comments that led to improve the quality of the manuscript. In the revised manuscript, we have done our best to address all raised comments. Responses to the technical comments are listed below and the corresponding changes have been applied to the revised manuscript

Round  2

Reviewer 1 Report

The paper has gone through a major revision and included deep learning based methods as mentioned in the previous review comments. I guess now the paper contains enough material as a survey paper.

Author Response

Degraded Historical document Binarization: A review on issues, challenges, techniques and future directions

Journal of Imaging

March 31, 2018

Dear editor and respected reviewers,

 We sincerely would like to thank the respected anonymous reviewers for their constructive and precise comments that led to improve the quality of the manuscript. In the revised manuscript, we have done our best to address all raised comments regarding English grammar issues and upper/lower cases so a precise proofreading has done for the revised paper.

Reviewer 2 Report

After the first review, the paper is in much more quality to be published. However, the language and wording style still requires extensive proofreading. There are many grammar issues and upper/lower case inconsistencies to be ready for publication. That is why I recommend authors carefully review these matters before submitting the paper again.

Author Response

Degraded Historical document Binarization: A review on issues, challenges, techniques and future directions

Journal of Imaging

March 31, 2018

Dear editor and respected reviewers,

 We sincerely would like to thank the respected anonymous reviewers for their constructive and precise comments that led to improve the quality of the manuscript. In the revised manuscript, we have done our best to address all raised comments regarding English grammar issues and upper/lower cases so a precise proofreading has done for the revised paper.

J. Imaging EISSN 2313-433X Published by MDPI AG, Basel, Switzerland RSS E-Mail Table of Contents Alert
Back to Top